# Levels of 25-hydroxy Vitamin D3 and Vitamin D Receptor Polymorphism in Severe Dengue Cases from New Delhi

**DOI:** 10.3390/tropicalmed5020072

**Published:** 2020-05-03

**Authors:** Anita Chakravarti, Tanisha Bharara, Neeru Kapoor, Anzar Ashraf

**Affiliations:** 1Department of Microbiology, Shree Guru Gobind Singh Tricentenary University, Gurugram (Haryana) 1222505, India; tanishabharara.med@gmail.com; 2Department of Microbiology, Maulana Azad Medical College, New Delhi 110002, India; drnr.kapoor@gmail.com (N.K.); anzar.ashraf@gmail.com (A.A.)

**Keywords:** dengue, severe dengue, primary dengue, secondary dengue, immunomodulator, 25-hydroxy vitamin D3, vitamin D receptor, restriction fragment length polymorphism, electrochemiluminescence assay, gene polymorphism

## Abstract

Background: Dengue is the “phoenix” that never went to ashes. First identified in 1943, in Japan, dengue virus has worldwide distribution and is a grave public health concern in developing countries like India; Methods: A cross sectional study was conducted among adults suspected of having dengue fever and attending Lok Nayak Hospital, New Delhi. Restriction Fragment Length Polymorphism was completed for the detection of vitamin D receptor (VDR) gene polymorphism; Results: Serum 25-hydroxy vitamin D3 (vitamin D) levels were found to be 1.6 times elevated in severe dengue cases as compared to healthy controls. Vitamin D levels were significantly higher in secondary infections compared to primary infections as well as secondary severe dengue cases as compared to secondary non-severe cases (*p* value < 0.05). A significant association of the T allele (rs2228570) was seen in severe dengue cases, while, when comparing the A/A with A/C and C/C genotypes (rs7975232) among dengue cases and healthy controls, the odds ratio was estimated to be 1.24 (0.55–2.75, *p* > 0.05) and 0.28 (0.08–0.96, *p* < 0.05) respectively; Conclusions: The present study is an attempt at decoding the role of vitamin D in dengue disease pathogenesis and exploring the role of genetic polymorphism in dengue disease pathogenesis.

## 1. Introduction

It is estimated that 2.5–3 billion of the world’s population are at risk of dengue infection [1]. The escalating magnitude of the problem, together with the changing epidemiology of dengue, is a serious public health concern [2]. 25-hydroxy vitamin D3 (Vitamin D) is known to play a key role in calcium homeostasis. Although its role as an immunomodulator has been known for decades, it is only recently that its therapeutic/prognostic role is being studied widely [3,4,5,6,7,8,9,10]. Its active hormonal form (1, 25-hydroxy vitamin D3) binds to vitamin D receptor (VDR) and translocates to nuclei to influence gene expression [4]. The vitamin D receptor is coded by the VDR gene on chromosome 12. It is a transcription factor and member of the steroid hormone nuclear receptor family. Several single nucleotide polymorphisms (SNPs), namely, BsmI, ApaI, TaqI and FokI, are known to influence the activity of VDR [2]. These SNPs affect the translation initiation region, consequently affecting the transcriptional activity of the vitamin D receptor [11,12]. Genetic variations in the VDR gene have been shown to be associated with infectious diseases, non-communicable diseases as well as various carcinomas like ovarian cancer [13,14,15,16]. Few studies have shown the association of the vitamin D receptor gene polymorphisms with susceptibility towards dengue virus infection [17,18].

Adequate levels of vitamin D promote innate immunity, while it exerts an inhibitory effect on the adaptive immune system [19]. Vitamin D has been reported to influence the expression of dengue virus entry receptor, dendritic cell specific intercellular adhesion molecule-grabbing non integrin (DC-SIGN) and FCγRIIA in immune cells [20,21,22]. A study reported higher levels of vitamin D in dengue virus infected patients compared to healthy controls [23]. However, the role of vitamin D serum level in dengue pathogenesis and the association of VDR gene polymorphisms with the clinical severity of dengue infection has not been extensively studied in an Indian context. In view of the above, this study was undertaken as one of the few published reports investigating the levels of vitamin D and genetic variations in vitamin D receptors with the dengue virus infection.

## 2. Materials and Methods

A cross sectional study was conducted among adults suspected of having dengue fever, visiting Lok Nayak Hospital, New Delhi. The study was conducted in the virology laboratory, Department of Microbiology, Maulana Azad Medical College, New Delhi. We studied 100 cases (suspected dengue patients) and 100 healthy controls (any subject without fever within last 3 months) over a period of 1 year (January 2014–December 2014). Peripheral venous samples were collected aseptically and serum separated. Serum was stored at −70 °C until further processing. Dengue was diagnosed by testing serum samples for NS1 antigen, dengue IgM and IgG antibody (depending upon the day of fever). Detection of NS1 antigen and dengue IgM antibody was undertaken using Panbio Dengue Early ELISA kits, Inverness Medical Innovations Australia (catalogue numbers: 01PE40 and E-DEN01M) for NS1 and dengue IgM respectively. Dengue IgG antibody was detected using Panbio Dengue IgG capture ELISA kit, Inverness Medical Innovations Australia (E-DEN02G). Primary and secondary dengue infection was distinguished on the basis IgG Panbio Dengue Early ELISA kit. A positive result (>22 Panbio units) was indicative of an active secondary infection (as per kit literature). Serum 25-hydroxy vitamin D3 (vitamin D) level was estimated using Electrochemiluminescence assay (ELECSYS 2010; Roche Diagnostics GmbH, Mannheim, catalogue number: 07464215190). Samples were processed and results interpreted according to kit literature. Restriction Fragment Length Polymorphism was undertaken for the detection of VDR gene polymorphism (start codon rs2228570 and 3′ UTR, rs7975232) using the methodology of Pani et al. [24]. Genomic DNA was extracted from patients and healthy controls using commercially available kit (QIAGEN) according to the manufacturer’s instructions. Extracted DNA was used for amplification of desired PCR product. The PCR product was digested with restriction enzyme FokI (Thermo Fisher, FD2144) and ApaI (Thermo Fisher, FD1414), for rs2228570 and rs7975232 polymorphism respectively after incubation at 37 °C for 9 h. The digested product was loaded onto 3% nusieve agarose gel and the restriction pattern analyzed using molecular weight marker ϕX174/HaeIII (MBI Fermentas, Lithunia) and Gel-Doc System (Alpha Innotech, San Leandro, USA).

### 2.1. Ethics Statement

The investigations were carried out following the rules of the Declaration of Helsinki of 1975, revised in 2013. The study was conducted after approval by the Institution’s ethics committee, Maulana Azad Medical College, 2013 (F.2/IEC/MAMC/11/No.52). All patients were included in the study after taking a written informed consent. Confidentiality of subjects was maintained throughout the study.

### 2.2. Inclusion Criteria for a Case of Dengue

Revised WHO guidelines (2009) were followed for inclusion of a case of dengue and its characterization [25].

### 2.3. Exclusion Criteria for a Case of Dengue

All cases with negative serology for dengue fever were excluded from the study.

### 2.4. Statistical Analysis

Data were analyzed using SPSS software (version 17.0). Continuous variables were presented as median (range) or mean ± SD, whereas categorical variables were expressed as frequencies (%). Differences between continuous variables were assessed by Student’s t-test, whereas categorical variables were evaluated using Pearson’s x^2^ test. The Hardy–Weinberg equilibrium for VDR genotypes was performed by x^2^ test. Allelic and genotypic frequency difference between study and control groups was estimated by 2 × 2 contingency table and Fisher exact test. *p* value < 0.05 was considered statistically significant.

## 3. Results

Out of 100 clinically suspected dengue patients, 43 were dengue positive. Among these, five were anti-dengue IgM positive, 25 were dengue NS1 antigen positive and 13 were positive for both. All the healthy controls (100) were serologically negative for dengue.

Males were predominant among the cases, comprising 62.8%, while females were 37.2%. The age of the cases ranged from 18 to 60 years, with 37.2% belonging to the age group of 21–30 years. Most of the cases came from Delhi and National Capital Region (65.12% from Delhi, 18.10% from Uttar Pradesh and 6.97% from Haryana), while 4.65% patients belonged to Bihar and Punjab (all these states comprise the catchment area of Lok Nayak and associated hospitals). Out of the 43 dengue positive patients, 28 were dengue cases with/without warning signs, whereas 15 were severe dengue cases. Twenty-five cases were primary infection, while 18 cases were secondary infections.

Fever was the most commonly presented symptom among dengue positive cases (100%) followed by retro orbital pain (86%), headache (76.7%) and arthralgia (67.44%). Petechiae were the most common signs (62.8%). The platelet count was significantly lower in severe dengue patients when compared to dengue patients, while hematocrit was significantly raised in severe dengue patients (*p* < 0.05). The liver function test was deranged (elevated ALT, AST, ALP and bilirubin levels) in severe dengue patients Table 1.

Vitamin D levels were investigated in 43 dengue positive cases and 100 healthy controls. Vitamin D levels were found to be higher in dengue cases (42.5 ± 14.2 nmol/L) as compared to healthy controls (33.8 ± 10.2 nmol/L). Vitamin D levels were further investigated in 28 non-severe dengue cases, 15 severe dengue cases and 100 healthy controls Table 2.

The levels were found to be 1.26 and 1.6 times elevated in non-severe cases (36.6 ± 8.7, CI 33.4–39.8) and severe dengue cases (53.5 ± 16.3, CI 45.2–61.7) respectively when compared to healthy controls (33.8 ± 10.2, CI 31.8–35.8), as shown in Table 2. Vitamin D levels were significantly higher in secondary infections (53.5 ± 13.8, CI 47.1–59.9) compared to primary infections (34.6 ± 8.1, CI 31.1–38.1) (1.5 times higher, *p* < 0.00001). Additionally, secondary severe dengue cases had levels significantly higher than secondary non-severe cases (*p* < 0.00001), as shown in Table 2. 

VDR gene polymorphism (rs 2228570) was screened in the samples and three different types of genotypes, i.e., C/C, C/T and T/T were found. The genotypes C/C, C/T and T/T were found in 11 (25.6%), 18 (41.9%) and 14 (32.6%) dengue positive cases and 59%, 25% and 16% healthy controls, respectively. Compared to the C/C genotype in healthy controls and dengue positive cases, it was estimated that the Odds Ratios (OR) of 3.86 (1.59–9.35, *p* = 0.002) and 4.69 (1.79–12.3, *p* = 0.001) for C/T and T/T frequency of genotypes were significantly higher in dengue cases. Another VDR gene polymorphism (rs 7975232) was screened in dengue positive cases and healthy controls. Three different genotypes A/A, A/C and C/C were found in 34.9%, 55.8%, 9.3% dengue positive cases and 31%, 40%, 29% healthy controls respectively. Compared to the A/A genotype in healthy controls and dengue positive cases, it was estimated that OR of 1.24 (0.55–2.75, *p* > 0.05) and 0.28 (0.08–0.96, *p* < 0.05) for A/C and C/C frequencies of genotypes. The lower frequency of C/C was found to be significant (*p* = 0.032), as shown in Table 3 and Figure 1.

When compared with dengue non-severe and severe cases, the frequency of the T allele of rs2228570 polymorphism and C allele of rs 7975232 in a dominant mode of inheritance (C/T + T/T genotype and A/C + C/C genotypes) was not found to be significantly higher (*p* > 0.05), as shown in Table 4.

## 4. Discussion

The majority of dengue cases were reported during the months of July and August. This could be attributed to the ideal climatic conditions for vector breeding (heavy rainfall—236 mm and 248 mm and increased humidity—60% and 64% in the months of July and August, respectively) [4]. In an eco-epidemiological analysis, the authors emphasize the importance of climatic factors such as rainfall, temperature and relative humidity as major determinants for dengue [26]. The sex-wise distribution of dengue cases demonstrated male predominance. Chakravarti et al. reviewed the epidemiology of dengue in India over the last fifty years. The authors reported that males outnumbered females in the majority of the reports of dengue outbreaks [27]. The finding can be explained by greater mobility of males, for the purpose of education/employment/social obligations, in Indian society.

We diagnosed 43 dengue positive cases in our study. Among the dengue positive cases, 65.1% were non-severe cases while 34.9% were severe dengue. In another study conducted in our center, the total dengue cases diagnosed over a period of one year was 56. Out of these, 67.85% were non-severe dengue and 32.15% were severe dengue [28].

The serum vitamin D level was measured among non-severe dengue, severe dengue cases and apparently healthy controls. The levels were found to be 1.26 times higher among dengue cases as compared to controls. Among the dengue cases, vitamin D levels were found to be 1.6 times elevated in severe cases as compared to non-severe dengue cases. Further vitamin D levels were 1.5 times higher among secondary infection when compared to primary infections. Secondary severe dengue infections had levels significantly higher than secondary non-severe infections (*p* = 0.003), while vitamin D concentrations were not significantly different among primary severe and non-severe infections (*p* = 0.778). The confidence intervals for means were calculated. A few of the confidence intervals are relatively large, as depicted in Table 2. Therefore, further studies, with a larger sample size, are vital. In a study conducted by Alagarasu et al., vitamin D concentrations were found to be significantly higher in dengue fever and dengue hemorrhagic fever cases as compared to healthy controls (*p* < 0.05). When the patients were grouped based on immune status and disease severity, secondary DHF cases displayed significantly higher concentrations of vitamin D as compared to secondary DF cases (*p* < 0.050) [23].

The development of severe dengue is hallmarked by a “cytokine storm” as depicted in Figure 2.

It is believed that a combined effect of three factors, namely antibody-enhancement (ADE), T cell response and shift from TH1 response to TH2 response, lead to the genesis of this phenomenon resulting in the development of DHF. Vitamin D is known to inhibit the T-helper1 (Th1) response and enhance the IL-10 response as depicted in Figure 1. A study conducted in New Delhi, India, demonstrated a significant rise in serum IL-10 levels in severe dengue infection as compared to healthy controls and non-severe dengue fever [28]. This might explain the association of higher vitamin D levels with severe dengue cases in our study, as shown in Figure 1. In contrast to the above findings, in an in vitro study (mosquito C6/36HT cell line) conducted by Alzate et al., at University of North Carolina, United States, vitamin D was found to restrict dengue virus (DENV) infection in human monocyte-derived macrophages by affecting DENV binding to cells [22]. This was an in vitro study; in vivo dengue disease outcome is determined by a multifaceted interaction between host- and pathogen-related factors. Viral virulence, host genetics and host immune response are the major protagonists of this fascinating interplay. This might explain the difference in findings. Another study conducted by Giraldo et al. reported high-dose vitamin D supplementation to be associated with reduced susceptibility of monocyte derived macrophages to dengue virus infection and pro-inflammatory cytokine production [29]. In contrast, our study group consisted of dengue patients who did not receive any vitamin D supplements, since we wanted to assess the role of existing level of vitamin D in these patients.

Vitamin D has been reported to influence the expression of DENV receptors on immune cells [20,21,22]. A few studies have explored the role of Vitamin D receptor gene polymorphism in dengue virus infection [17,18]. In the present study, the start codon polymorphism (rs2228570) and 3′UTR polymorphism (rs7975232) were investigated among dengue patients and healthy controls. Results revealed a significantly higher frequency of C/T + T/T genotypes of rs2228570 polymorphism. This polymorphism was observed in severe dengue cases, suggesting an association of the T allele with disease severity. In a study conducted in Vietnam, Loke et al. emphasized the association of the T allele with dengue disease severity [17]. The authors attributed the findings to a weaker TH1 immune response and thus reduced cellular immunity among these patients. Based on the frequency of the minor allele of rs7975232, this study showed a lower frequency of the C allele in all dengue cases requiring hospitalization as compared to apparently healthy controls. This might suggest that the C/C genotype of rs7975232 is associated with a reduced risk of developing symptomatic dengue.

## 5. Conclusions

Vitamin D levels were 1.6 times elevated among severe dengue cases as compared to the controls. Further correlation of vitamin D levels with pro-inflammatory/anti-inflammatory cytokines might create a way to establish its role as a cofactor for the prediction of disease severity. An increasing number of defined polymorphisms in the human genome will boost the potential of genetic susceptibility studies in dengue, paving the way for the development of possible therapeutic options. Furthermore, carefully premeditated multi-country randomized controlled trials encompassing large populations of diverse race, ethnicity and genetics is required to further fill the knowledge gaps in the role of vitamin D on dengue disease outcome.

## Figures and Tables

**Figure 1 tropicalmed-05-00072-f001:**
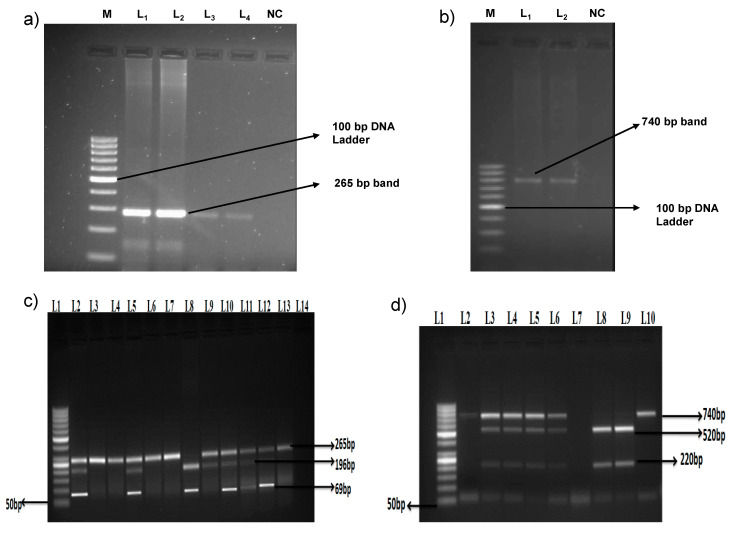
Amplification and genotyping of VDR gene polymorphisms (rs2228570 and rs7975232) (**a**) Amplification of VDR rs2228570 polymorphism; M—Molecular marker 100bp, Sample—L_1_ and L_2_, L_3_ and L_4_ and NC—Negative control (**b**) Amplification of VDR rs7975232 polymorphism; M—Molecular marker 100bp, Sample—L_1_ and L_2_, NC—Negative control (**c**) Genotyping of VDR gene polymorphism (rs2228570); L_1_—Molecular marker 50bp and Sample—L_2_- L_14_ (**d**) Genotyping of VDR gene polymorphism (rs7975232); L_1_—Molecular marker 50bp and Sample—L_2_- L_10_.

**Figure 2 tropicalmed-05-00072-f002:**
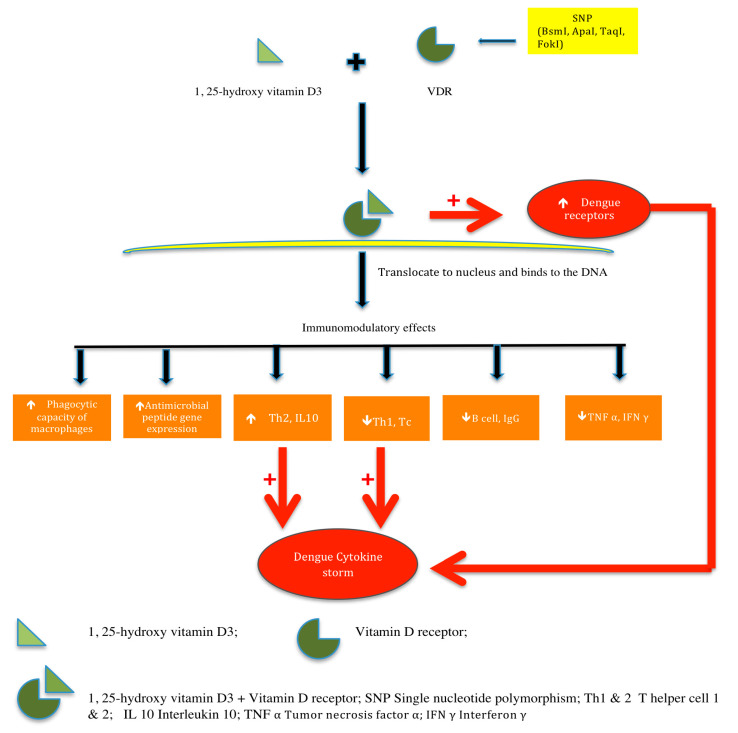
Immunomodulatory effects of vitamin D and its possible role in dengue pathogenesis.

**Table 1 tropicalmed-05-00072-t001:** Distribution of clinical symptoms and signs in study group (n = 43).

**Clinical Symptoms**	**n**
Fever	43
Headache	33
Retro orbital pain	37
Arthralgia	29
Myalgia	27
**Clinical Sign**	**n**
Rash	9
Malena	10
Petechiae	27
Gum bleeding	2
Epistaxis	4
Hematuria	4
Hepatomegaly	1
Splenomegaly	1

**Table 2 tropicalmed-05-00072-t002:** A comparative analysis of vitamin D levels in various study groups.

Groups (n)	Vitamin D Levels (nmol/L) Mean ± SD	95% CI	*p* Value
Controls (100)	33.8 ± 10.2	31.8–35.8	-
Cases (43)	42.5 ± 14.2	38.3–46.7	0.002
Non-severe (28)	36.6 ± 8.7	33.4–39.8	0.037
Severe (15)	53.5 ± 16.3	45.2–61.7	<0.0001
Primary (25)	34.6 ± 8.9	31.1–38.1	<0.0001
Secondary (18)	53.5 ± 13.8	47.1–59.9	
Primary non-Severe (20)	34.2 ± 8.8	30.3–38	0.778
Primary Severe (5)	42.6 ± 4.9	38.3–46.9	
Secondary Non-severe (8)	35.5 ± 5.1	32–39	0.003
Secondary Severe (10)	60.0 ± 13.6	59.1–60.9	

**Table 3 tropicalmed-05-00072-t003:** Distribution of vitamin D receptor (VDR) gene polymorphism (rs2228570 and rs7975232) in healthy controls and dengue positive cases.

Genotypes	Controls (100) n (%)	Cases (43) n (%)	OR (95% CI)	*p* Value
CC	59 (59)	11 (25.6)	1.0 (Ref)	
CT	25 (25)	18 (41.9)	3.86 (1.59–9.35)	0.002
TT	16 (16)	14 (32.6)	4.69 (1.79–12.3)	0.001
AA	31 (31)	15 (34.9)	1.0 (Ref)	0.032
AC	40 (40)	24 (55.8)	1.24 (0.55–2.75)	
CC	29 (29)	4 (9.3)	0.28 (0.08–0.96)	

**Table 4 tropicalmed-05-00072-t004:** Percentage genotype frequencies of VDR gene polymorphism (rs2228570 and rs7975232) in dengue non-severe and severe cases.

Genotypes	Non-Severe *n (%)	Severe *n (%)	OR (95% CI)	#*p* Value
AA	7 (58.3)	5 (41.7)	1.0 (Ref)	-
AC	17 (70.8)	7 (29.2)	0.57 (0.13–2.44)	0.453
CC	4 (57.1)	3 (42.9)	1.05 (0.15–6.92)	0.959
CC	12 (57.1)	9 (42.9)	1.0 (Ref)	-
CT	14 (82.4)	3 (17.6)	0.28 (0.06–1.30)	0.09
TT	2 (65.1)	3 (34.9)	2.00 (0.27–14.58)	0.48

*n = number of cases, #*p* value was calculated using Chi-Square test; significant at *p* < 0.05.

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
