# Peer review of "Levels of 25-hydroxy Vitamin D3 and Vitamin D Receptor Polymorphism in Severe Dengue Cases from New Delhi"

_tropicalmed, 2020, doi:10.3390/tropicalmed5020072_

Round 1

Reviewer 1 Report

Reviewer:  The manuscript from Anita Chakravarti et al. describes results of a study where 43 Dengue virus patients were analyzed for 25-hydroxy vitamin D3 levels and vitamin D receptor polymorphism. The aim was to find correlates between negative controls, non-severe and severe Dengue cases. They showed that in 15 severe Dengue cases a correlation was found.

Overall the methodology of the study was performed adequately, but unfortunately, the results and their analysis are presented in a highly confusing way.

The discussion does explain the results and relevant literature is cited. However, it would be helpful to put Fig 1 into the discussion section since this figure is cited but unfortunately not considered or explained in the introduction section.

In general, figures should be clearly legible after b/w printing and compounds used or analyzed should be named consistently. The authors should kindly follow the recommendations and mdpi guidelines for authors for the preparation of the manuscript and figures. Altogether, presentation of data in figures and tables is inadequate.

The authors should consider the following points to improve their manuscript:

Major points:

Reviewer:   Title of the manuscript is too long and implies a role in Dengue disease in general. Since the study group that shows a positive correlation is a small study group it is more like a case report and the title should be revised. For example: Levels of 25-hydroxy vitamin B3 and Vitamin D receptor polymorphism in severe Dengue cases from New Dehli.

Reviewer: Lane 51: Figure 1 should be placed in the discussion section to further explain the results. The figure needs a legend that describes the parts and abbreviations within the figure. Within the figure it would be wise to use only one description for each compound used or analyzed. For example, a different description for 25-hydroxy vitamin D3 is present in the whole manuscript incl. tables and this figure also. The VDR abbreviation should not be explained in the diagram, but in the legend, at the first appearance in the figure legends. What is the reason for the exclamation mark (!), extensively used in the diagram?

Reviewer:   Lane 125, Table 1: in the table, n is the proportion of cases with the clinical symptoms as listed. It is not = 43. Also, frequencies in % by such a small number of cases are more confusing than helpful. For example Fever 43/43, or Myaligia 27 of 43 etc.

Reviewer: Lane 128: Table 2:  Values for 25-hydroxy vitamin D3 level vary by +/-  ¼, thus the statement that levels “found to be  significantly higher in dengue cases” is clearly not supported by the data.

There is a significant difference between the severe and the 128 cases of non-severe and negative cases. p Values are not applicable when study groups are compared with such a high difference in participants. Especially when the 25(OH)D3 low level group is 8-9 times larger compared to the group of severe Dengue. Additionally, for a better explanation of the results it would be helpful to specify the number of cases for each group listed in the table.

Reviewer: Lane 134: Table 3: Data should be combined with table 2. Numbers of cases for each group should be specified in the table. The indication of percentages is not necessary. Please see also my comment for lane 125.

Reviewer:  Lane 146: Fig. 2:  The labels for the x- and y-axis are missing. The figure must be legible after b/w printing. The figure needs an informative legend that explains its content. The figure can also be switched into a table like table 4, since they transport an similar message.

Reviewer:  Lane 152: Tab 4:   The citation of table 4 in the text at lane 151 refers to data not presented in the table. This section 149-154 needs a complete revision to make it clear and understandable to the reader.

Reviewer:  Lane 162: Fig. 3. This section 155-161, also needs a complete revision to make it clear and understandable to the reader. In addition, the figure needs a legend explaining the lane labels in 3a-d.  Labels for the DNA marker are missing. Fragment sizes must be explained. Markings should be consistent in font and size. Font size in c and d is much too small.

Minor points:          

Reviewer: Lane 189: What is the message of Pune et al that is in accordance? The authors should compare results and not just cite the literature.

Reviewer: Lane 203: What are similar findings? The authors should compare results and not just cite the literature.

Reviewer:  Lane 211: Authors are writing about the tetravalent vaccine Dengvaxia which has nothing to do with the topic and main results of the paper nor is the Dengue pathogenesis an enigma. The conclusion can be deleted or should refer to the small study group, the 1.6 enhanced 25-hydroxy vitamin D3 levels, and the perspective or scientific work that has to be done in the future.

Reviewer: The authors claim that their results can be used as a prognostic marker for severe dengue. Since the study group is small such a prediction is highly questionable. However, how will this be put into operation? Are there prognostic examples?

Reviewer:  I suggest that the authors rethink their statistical analysis since it is clearly overstated for a group of 15 severe cases. Two digits behind the comma? Is this the accuracy of the study? I guess not.

Author Response

1.

Title of the manuscript is too long and implies a role in Dengue disease in general. Since the study group that shows a positive correlation is a small study group it is more like a case report and the title should be revised. For example: Levels of 25-hydroxy vitamin D3 and Vitamin D receptor polymorphism in severe Dengue cases from New Delhi.

Thank You for the suggestion. The title has been modified.

Page 1; Line 2-3

2.

Lane 51: Figure 1 should be placed in the discussion section to further explain the results. The figure needs a legend that describes the parts and abbreviations within the figure. Within the figure it would be wise to use only one description for each compound used or analyzed. For example, a different description for 25-hydroxy vitamin D3 is present in the whole manuscript incl. tables and this figure also. The VDR abbreviation should not be explained in the diagram, but in the legend, at the first appearance in the figure legends. What is the reason for the exclamation mark (!), extensively used in the diagram?

Figure shifted to discussion section.

All the necessary changes made in figure.

Page 6-7; Figure 2

3.

Lane 125, Table 1: in the table, n is the proportion of cases with the clinical symptoms as listed. It is not = 43. Also, frequencies in % by such a small number of cases are more confusing than helpful. For example Fever 43/43, or Myaligia 27 of 43 etc.

All the 43 cases presented with one or more sign/symptom of dengue. This is the reason n=43. Percentages have been removed from the table. All the suggested changes have been made in table 1

Page 3; Line 110

4.

Lane 128: Table 2:  Values for 25-hydroxy vitamin D3 level vary by +/-  ¼, thus the statement that levels “found to be  significantly higher in dengue cases” is clearly not supported by the data.

The statement has been changed as per suggestion.

Page 3; Line 111-113

5.

There is a significant difference between the severe and the 128 cases of non-severe and negative cases. p Values are not applicable when study groups are compared with such a high difference in participants. Especially when the 25(OH)D3 low level group is 8-9 times larger compared to the group of severe Dengue. Additionally, for a better explanation of the results it would be helpful to specify the number of cases for each group listed in the table.

The number of cases for each group has been specified in the table. p value has been removed

Page 3-4;Table 2

6.

Lane 134: Table 3: Data should be combined with table 2. Numbers of cases for each group should be specified in the table. The indication of percentages is not necessary. Please see also my comment for lane 125.

Thank you for the suggestion. Table 2 and 3 have been combined and cases in each group specified.

Page 3-4;Table 2

7.

Lane 146: Fig. 2:  The labels for the x- and y-axis are missing. The figure must be legible after b/w printing. The figure needs an informative legend that explains its content. The figure can also be switched into a table like table 4, since they transport an similar message.

The figure has been changed into a table

Page 4; Table 3

8.

Lane 152: Tab 4:   The citation of table 4 in the text at lane 151 refers to data not presented in the table. This section 149-154 needs a complete revision to make it clear and understandable to the reader.

Section modified and re-written

Page 4; 122-137

9.

Lane 162: Fig. 3. This section 155-161, also needs a complete revision to make it clear and understandable to the reader. In addition, the figure needs a legend explaining the lane labels in 3a-d.  Labels for the DNA marker are missing. Fragment sizes must be explained. Markings should be consistent in font and size. Font size in c and d is much too small.

-Section revised and re-written.

-Figure has been modified.

-Page 5;122-137

-Page 5; Figure 1

10.

Lane 189: What is the message of Pune et al that is in accordance? The authors should compare results and not just cite the literature.

The findings have been elaborated

Page 6; 167-171

11.

Lane 203: What are similar findings? The authors should compare results and not just cite the literature.

The study findings relevant to our study have been added

Page 7; 197-199

12.

Lane 211: Authors are writing about the tetravalent vaccine Dengvaxia which has nothing to do with the topic and main results of the paper nor is the Dengue pathogenesis an enigma. The conclusion can be deleted or should refer to the small study group, the 1.6 enhanced 25-hydroxy vitamin D3 levels, and the perspective or scientific work that has to be done in the future.

Conclusion has been modified

Page 7; 204-210

13.

The authors claim that their results can be used as a prognostic marker for severe dengue. Since the study group is small such a prediction is highly questionable. However, how will this be put into operation? Are there prognostic examples?

Currently four classes of prognostic markers have been defined for dengue, molecular markers (genetic); immunological markers (cytokines); endothelial activation factors and biochemical markers. Further correlation of vitamin D levels with pro-inflammatory/anti-inflammatory cytokines might pave a way to establish its role as cofactors for prediction of disease severity.

14.

I suggest that the authors rethink their statistical analysis since it is clearly overstated for a group of 15 severe cases. Two digits behind the comma? Is this the accuracy of the study? I guess not.

Thank You for the suggestion. Changes have been made in all the relevant tables and text.

Throughout the manuscript

Reviewer 2 Report

  1. In my opinion, the word “role” in the title should be changed. It may be better to remove or change for another word like association.
  2. The introduction should include previous reports that evaluated vitamin D3 levels and vitamin D receptor polymorphisms in dengue.
  3. Could you provide more information about the kits used for dengue diagnosis? In the text, the authors just refer to Pambio Dengue Early ELISA kit, which is for NS1 detection. I understand that the other kits may be from Pambio as well, but could you also provide catalog numbers? The same should be done for the determination of vitamin D3 levels as well as the genotyping of VDR gene polymorphisms.
  4. Line 109, how many of these patients were positive for IgG alone, IgM/IgG or IgG/NS1? I know that IgG was used to classify as primary or secondary but I think it would be interesting seeing this data.
  5. A concern is that the results may not be conclusive due to the low number of dengue subjects included in each dengue group comparison (for example data from table 3).
  6. In tables 2 & 3, the n= of patients can be added to each group.
  7. I would suggest including data from all dengue subjects in table 2 as it is referred to in the text.
  8. Line 133, It may be a mistake referring the table 3.
  9. Could you please describe better the legend of figure 3, please?
  10. Overall, the discussion could be improved as there are many studies covering vitamin D3 in dengue. Until now, the discussion covers mostly the result section again.

Author Response

1.

In my opinion, the word “role” in the title should be changed. It may be better to remove or change for another word like association.

Thank you for the suggestion. We have modified the title

Page 1; 2-3

2.

The introduction should include previous reports that evaluated vitamin D3 levels and vitamin D receptor polymorphisms in dengue.

Introduction has been modified

Page 2; 44-53

3.

Could you provide more information about the kits used for dengue diagnosis? In the text, the authors just refer to Pambio Dengue Early ELISA kit, which is for NS1 detection. I understand that the other kits may be from Pambio as well, but could you also provide catalog numbers? The same should be done for the determination of vitamin D3 levels as well as the genotyping of VDR gene polymorphisms.

Relevant information has been added.

Page 2;61-74

4.

Line 109, how many of these patients were positive for IgG alone, IgM/IgG or IgG/NS1? I know that IgG was used to classify as primary or secondary but I think it would be interesting seeing this data.

-IgG alone positive=0

-IgM/IgG=4

-IgG/NS1=1

5.

A concern is that the results may not be conclusive due to the low number of dengue subjects included in each dengue group comparison (for example data from table 3).

Changes have been made as suggested by the other reviewer. p value has been removed, only difference in vitamin D levels is discussed.

Page 4;table 3

6.

In tables 2 & 3, the n= of patients can be added to each group.

Tables have been merged. Suggestion taken into consideration in table 2.

Page 3-4; table 2

7.

I would suggest including data from all dengue subjects in table 2 as it is referred to in the text.

Section modified and re-written

Page 4; 122-137

8.

Line 133, It may be a mistake referring the table 3.

Data has been incorporated in table 2

Page 3-4; table 2

9.

Could you please describe better the legend of figure 3, please?

Change has been made in figure legend

Page 5; Figure 1

10.

Overall, the discussion could be improved as there are many studies covering vitamin D3 in dengue. Until now, the discussion covers mostly the result section again.

The discussion has been modified

Page 6-7

Round 2

Reviewer 1 Report

Reviewer:  The manuscript from Anita Chakravarti et al. was resubmitted and the modified version shows improvements.

The authors should consider the following points to finalize their manuscript:

Major points:

Reviewer:   The legend for Fig. 2 is missing.

Reviewer:   Title of Fig. 1 is missing. Do not start with (a).

Reviewer:   Coming back to statistical analysis. You have a small sample size and therefore your p-values and/or confidence interval surrounding your statistical test results will tell you how much you can generalize your results to the whole population as it will tell you the likely range that the result will take in the total population with, e.g. (if a 95% confidence interval), 95% confidence. The larger your sample, the closer your confidence interval and the more realistic your results are the better your statistical test of the null hypothesis will be. A sample size of 43 is quite small so I think that your confidence intervals will be relatively large. Thus, you won't be really sure of what the true population data are, assuming your group is representative of the population, and that’s the important part.

Hope this helps, and I would recommend to reconsider seriously your statistical analysis. However it would not change your D3 data but would show scientific correctness.

Minor points:

Reviewer:  Table 1 needs some tabs adjustment.

Reviewer:  Table 2 and 3 must be reformatted correctly.

Reviewer:   The manuscript is full of little mistakes and typing errors. Is your word processor short on spaces? See for example lines 41, 62, 64, 73, 74, 100, 115.

Author Response

1.

The legend for Fig. 2 is missing

The legend has been highlighted in red. All the parts and abbreviations have been explained below the figure.

Page 7; Line 177

2.

Title of Fig. 1 is missing. Do not start with (a).

Title has been added

Page 6; Line 143

3.

Small sample size and therefore your p-values and/or confidence interval surrounding your statistical test results will tell you how much you can generalize your results to the whole population as it will tell you the likely range that the result will take in the total population with, e.g. (if a 95% confidence interval), 95% confidence. The larger your sample, the closer your confidence interval and the more realistic your results are the better your statistical test of the null hypothesis will be. A sample size of 43 is quite small so I think that your confidence intervals will be relatively large. Thus, you won't be really sure of what the true population data are, assuming your group is representative of the population, and that’s the important part.

Hope this helps, and I would recommend to reconsider seriously your statistical analysis. However it would not change your D3 data but would show scientific correctness.

The 95% confidence intervals for means have been calculated. They range from small (1.9) to relatively large (16.5).

Page 3-4; Table 2

Page 7, Line 169-171

4.

Table 1 needs some tabs adjustment.

Changes have been made

Page 3; Table 1

5.

Table 2 and 3 must be reformatted correctly.

Changes have been made

Page 3-5;Table 2 & 3

6.

The manuscript is full of little mistakes and typing errors. Is your word processor short on spaces? See for example lines 41, 62, 64, 73, 74, 100, 115.

Manuscript reformatted
